



# Integrated Methane Inversion (IMI 1.0): A user-friendly, cloud-based facility for inferring high-resolution methane emissions from TROPOMI satellite observations

Daniel J. Varon[1], Daniel J. Jacob[1], Melissa Sulprizio[1], Lucas A. Estrada[1], William B. Downs[1], Lu Shen[2], Sarah E. Hancock[1], Hannah Nesser[1], Zhen Qu[1], Elise Penn[1], Zichong Chen[1], Xiao Lu[3], Alba Lorente[4], Ashutosh Tewari[5], Cynthia A. Randles[5]

[1]Harvard University, Cambridge, Massachusetts, United States
[2]Department of Atmospheric and Oceanic Sciences, School of Physics, Peking University, Beijing, China
[3]Sun Yat-Sen University, Guangzhou, China
[4]SRON Netherlands Institute for Space Research
[5]ExxonMobil Research and Engineering Company, Annandale, New Jersey, United States

*Correspondence to*: Daniel J. Varon (danielvaron@g.harvard.edu)

**Abstract.** We present a user-friendly, cloud-based facility for quantifying methane emissions with $0.25° \times 0.3125°$ ($\approx 25 \times 25$ km$^2$) resolution by inversion of satellite observations from the TROPOspheric Monitoring Instrument (TROPOMI). The facility is built on an Integrated Methane Inversion optimal estimation workflow (IMI 1.0) and supported for use on the Amazon Web Services (AWS) cloud. It exploits the GEOS-Chem chemical transport model and TROPOMI data already resident on AWS, thus avoiding cumbersome big-data download. Users select a region and period of interest, and the IMI returns an analytical solution for the Bayesian optimal estimate of emissions on the $0.25° \times 0.3125°$ grid including error statistics, information content, and visualization code for inspection of results. An out-of-the-box inversion with rectilinear grid and default prior emission estimates can be conducted with no significant learning curve. Users can also configure their inversions to infer emissions for irregular regions of interest, swap in their own prior emission inventories, and modify inversion parameters. Inversion ensembles can be generated at minimal additional cost once the Jacobian matrix for the analytical inversion has been constructed. A preview feature allows users to determine the TROPOMI information content for their region and time period of interest before actually performing the inversion. The IMI is heavily documented and is intended to be accessible by researchers and stakeholders with no expertise in inverse modelling or high-performance computing. We demonstrate the IMI's capabilities by applying it to estimate methane emissions from the US oil-producing Permian Basin in May 2018.



## 1 Introduction

Controlling methane emissions is a major focus of climate policy. Anthropogenic methane emissions are primarily from livestock, oil and gas operations, coal mining, waste management, and rice cultivation (Saunois et al., 2020). Emission inventories use "bottom-up" methods to estimate emissions from activity levels and emission factors in these different sectors,

but the emission factors are often highly uncertain (IPCC, 2019). "Top-down" inverse methods using satellite observations of atmospheric methane in combination with an atmospheric transport model and statistical optimization can evaluate the bottom-up inventories and monitor emissions worldwide, but they are difficult to use and have their own errors (Jacob et al., 2016).

Here we present an open-access, cloud-based facility for researchers and stakeholders to estimate methane emissions for user-selected regions of interest by performing high-resolution analytical inversions of TROPOMI satellite data archived

on the cloud, and including quality control and error characterization as part of the inversion results. This facility enables users to infer methane emissions from TROPOMI data without requiring expert knowledge of inverse methods or cumbersome data download. It exemplifies the emerging concept of "bringing compute to data" that is viewed as crucial for effective utilization of very large Earth science datasets (Yang et al., 2017).

Satellite instruments observe atmospheric methane column concentrations by solar backscatter in the shortwave

infrared (SWIR). Earlier instruments (SCIAMACHY, GOSAT) demonstrated effectiveness for inferring methane emissions on large regional scales (Bergamaschi et al., 2013; Wecht et al., 2014; Turner et al., 2015; Miller et al., 2019) but were limited by coarse pixel resolution (SCIAMACHY) or sparse sampling (GOSAT). The TROPOspheric Monitoring Instrument (TROPOMI), launched in October 2017 aboard the European Space Agency's Sentinel-5P satellite, offers unprecedented capability for monitoring emissions on regional scales, with daily global observations at $5.5 \times 7$ km$^2$ nadir pixel resolution

over land (Hu et al., 2018; Schneising et al., 2019; Lorente et al., 2021). The retrieval success rate averages only 3% because of clouds and dark/heterogeneous surfaces (Hasekamp et al., 2019) but the data density is still two orders of magnitude higher than for GOSAT (Qu et al., 2021). TROPOMI data have been used in regional inversions at up to 25-km resolution (Zhang et al., 2020; Shen et al., 2021, 2022).

Inversion of TROPOMI data to infer methane emissions requires a chemical transport model (CTM), known as

forward model for the inversion, to relate emissions to the observed methane columns through simulation of atmospheric transport. The problem is generally underconstrained because of uneven data density and because of errors in the satellite retrievals and in the CTM, referred to collectively as observational error. The solution must therefore be regularized, typically with prior information in the form of bottom-up emissions on the CTM grid, to produce posterior emission estimates that improve on the prior. This is generally done by minimization of a Bayesian cost function, using either variational methods or

an analytical solution (Brasseur and Jacob, 2017). Variational methods can infer methane emissions on any grid, for any nonlinear problem, and for any error probability density function (pdf), but they do not immediately provide error characterization of the posterior estimate. Analytical solution takes advantage of the linearity of the relationship between methane emissions and concentrations (Maasakkers et al., 2021). It requires explicit construction of the Jacobian matrix





expressing the sensitivity of concentrations to emissions, but this is readily done on supercomputing clusters as an
embarrassingly parallel problem (Maasakkers et al., 2019). Two major advantages of the analytical solution are that (1) it
provides closed-form characterizations of the posterior error pdf and the information content of the observations, and (2) it
allows easy generation of solution ensembles exploring the inversion parameter space (Lu et al., 2022).

Inverse analyses of satellite observations require complex modelling tools, advanced data processing, and access to
high-end computational resources. These are major barriers for novice and occasional users, and for stakeholders lacking
technical expertise. Our user-friendly, cloud-based facility for inferring high-resolution methane emissions from TROPOMI
satellite data lifts those barriers. The facility is based on an Integrated Methane Inversion workflow (IMI 1.0) that builds on
current best practices for analytical inversion of TROPOMI data (Shen et al., 2021). It draws on the GEOS-Chem CTM already
accessible on the Amazon Web Services (AWS) cloud (Zhuang et al., 2019; 2020), directly accesses the operational TROPOMI
data maintained on the cloud by Meteorological Environmental Earth Observation S.r.l. (MEEO), and infers methane
emissions at $0.25° \times 0.3125°$ ($\approx 25 \times 25$ km$^2$) resolution for user-selected regions. It is designed to be easily configurable for
users wishing to quantify emissions for specific regions and periods. The workflow can be run "out of the box" or modified
with user-supplied information, and it can be downloaded from the cloud for users wishing to work on their own computational
clusters. Our objective in this paper is to provide a high-level description of the facility. Detailed technical documentation for
user support is available online ([imi.seas.harvard.edu](imi.seas.harvard.edu)).

## 2 Integrated Methane Inversion (IMI)

The IMI infers methane emissions for a user-selected region and period by inversion of TROPOMI methane observations with
GEOS-Chem as forward model. The forward model $\boldsymbol{F}$ relates methane emissions (gridded state vector $\boldsymbol{x}$) to the observed
methane columns (observation vector $\boldsymbol{y}$) such that $\boldsymbol{y} = \boldsymbol{F}(\boldsymbol{x}) + \boldsymbol{\varepsilon_O}$, where the observational error $\boldsymbol{\varepsilon_O}$ includes errors in both
the satellite data and the forward model. The inversion optimizes $\boldsymbol{x}$ to match the observations, subject to constraints from the
prior emission estimates ($\boldsymbol{x_A}$), which have their own error $\boldsymbol{\varepsilon_A}$. The optimization is done by analytical minimization of a least-
squares Bayesian cost function, yielding a posterior estimate $\hat{\boldsymbol{x}}$ for the state vector with accompanying error statistics. Here we
describe the different components of the IMI and use a one-month inversion for the US Permian Basin (Fig. 1) as a guiding
example.

### 2.1 TROPOMI satellite observations

TROPOMI retrieves atmospheric methane columns from backscattered sunlight in the 2.3 µm methane absorption band, with
daily global coverage at $5.5 \times 7$ km$^2$ nadir pixel resolution ($7 \times 7$ km$^2$ prior to August 2019). Measurements are made at ~13:30
local solar time. The methane retrieval is produced by the Netherlands Institute for Space Research (SRON). It is based on the
RemoTeC full-physics algorithm (Butz et al., 2009; 2010; 2011) and retrieves methane data as column-average dry-air mixing
ratios $X_{CH4}$ (ppb) along with scattering properties of the atmosphere (Butz et al., 2012; Hu et al., 2016). The TROPOMI data



are posted operationally on the AWS cloud in a public S3 bucket and updated daily by MEEO with a latency of a few days
(https://registry.opendata.aws/sentinel5p). The methane product provides information on numerous retrieval parameters
together with $X_{CH4}$, including the centre and boundaries of the pixel; the surface pressure; the 12-layer pressure grid of the
retrieval; the vertical averaging kernel vector and prior vertical profile of methane dry-air mixing ratio; a quality assurance
value; and the retrieved surface albedo in the near infrared (NIR) and SWIR spectral ranges.

100         The operational TROPOMI record begins in May 2018. The methane retrieval is presently Version 1 (Hasekamp et
al., 2019) until July 2021 and Version 2 (Lorente et al., 2021) afterward. Validation of Version 1.03 showed a global mean
bias of -2.7 ppb relative to ground-based measurements from 19 sites in the Total Column Carbon Observing Network
(TCCON; Wunch et al., 2011a; Qu et al., 2021), but global bias is of no consequence for regional inversions because it is
effectively corrected through the boundary conditions. Of more concern are spatially variable biases (regional biases), caused

mainly by aliasing of surface albedo errors into the methane retrieval (Lorente et al., 2021). Qu et al. (2021) quantified a
nominal TROPOMI regional bias of 6.7 ppb in Version 1.03 as the standard deviation of station-to-station biases between
TROPOMI and the 19 TCCON sites, and a similar analysis for Version 2.02 shows a regional bias of 5.6 ppb (Lorente et al.,
2021). This is sufficiently small to enable successful regional inversions, for which Buchwitz et al. (2015) estimated a regional
bias threshold of 10 ppb. In the IMI we only use recommended high-quality retrievals over land, with quality assurance value

≥ 0.5 and with SWIR albedo < 0.05 (Hu et al., 2016). We further remove observations with high "blended albedo" (> 0.85), a
linear combination of NIR and SWIR albedo, to avoid biases from snow-covered scenes (Wunch et al., 2011b; Lorente et al.,
2021).

## 2.2 GEOS-Chem chemical transport model as forward model for the inversion

GEOS-Chem is a three-dimensional CTM that simulates methane concentrations on the basis of prescribed emissions either

globally or for user-selected nested domains (Wecht et al., 2014). It is driven by Goddard Earth Observation System (GEOS)
meteorological data from the NASA Global Modelling and Assimilation Office (GMAO). The IMI uses as default the GEOS
Fast Processing (GEOS-FP) meteorological data at 0.25° × 0.3125° resolution, with an option to use the GEOS Modern-Era
Retrospective Analysis for Research and Applications, version 2 (MERRA-2) at 0.5° × 0.625° resolution. The GEOS data
have 72 vertical levels from the surface to the mesopause, and these are condensed to 47 levels in our GEOS-Chem simulations

by merging levels in the upper stratosphere and mesosphere.

       We use the nested capability of GEOS-Chem to simulate methane concentrations over the inversion domain, with
dynamic boundary conditions outside the inversion domain updated every 3 hours from a global archive of TROPOMI data
smoothed spatially over a rolling ±10° window and temporally over a one-month period centred on each grid square and day,
and distributed vertically following a GEOS-Chem simulation at 4° × 5° resolution (Shen et al., 2021). This smoothed

TROPOMI 3-D archive is provided as part of the IMI. Using smoothed TROPOMI data as boundary conditions minimizes the
bias from boundary conditions advected over the user-selected region. Smoothing of the TROPOMI data is necessary because



of the sparsity of successful retrievals and the noise therein. To further reduce the bias associated with boundary conditions, we expand the inversion domain beyond the user-selected region of interest to include a buffer area, and coarse buffer elements are added to the state vector of methane emissions to be optimized (Figure 1; Section 2.3).

The user-specified period of interest defines the time window for the GEOS-Chem simulation. Starting from the smoothed TROPOMI fields as initial conditions, we apply a 1-month spin-up with prior emission estimates to properly initialize the model concentration fields within the inversion domain. One month is sufficient to fully ventilate any practical regional domain. This spin-up only needs to be done once.

The GEOS-Chem simulation includes chemical methane sinks from archived tropospheric concentrations of oxidants

(OH, Cl) and stratospheric loss frequencies (Maasakkers et al., 2019), as well as soil uptake (Murguia-Flores et al., 2018), but these are inconsequential for nested-domain simulations and are not optimized by the IMI. Ventilation of the inversion domain takes place on much shorter time scales than the methane atmospheric lifetime, and the sinks are relatively smooth, so no information on methane sinks is to be gained from a regional inversion. The effect of methane sinks is implicitly included in the specification of boundary conditions.


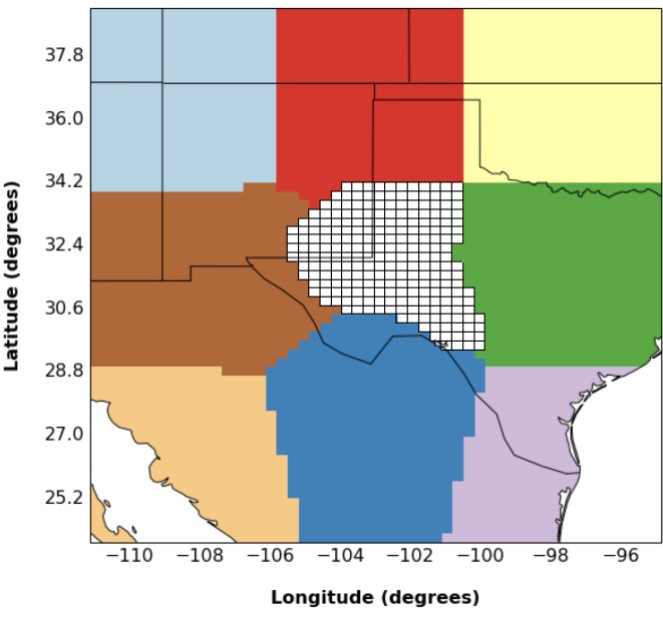

**Figure 1:** Example of an IMI state vector for inferring methane emissions from TROPOMI observations. Here the region of interest is the US Permian Basin in Texas and New Mexico (grid with white background), comprising 235 grid elements at 0.25° × 0.3125° resolution generated from a shape file. The inversion domain also includes the areas in colour bordering the region of interest, representing 8 buffer
elements added to the state vector to correct errors in boundary conditions (see Section 2.3).





### 2.3 Methane emission state vector to be optimized

The state vector $x$ is the ensemble of variables ("state variables") to be optimized in the inversion. In the IMI, these are the gridded methane emissions at 0.25° × 0.3125° resolution for the region of interest, plus buffer elements at coarser resolution

bordering the region of interest and filling out the inversion domain (8 elements by default). Users specify a region and time period of interest in the IMI configuration file. The region of interest can have any irregular shape, as illustrated in Figure 1. In that example case, the region of interest is an assemblage of 235 0.25° × 0.3125° grid cells covering the geological extent of the Permian Basin, and the 8 buffer elements expand to a rectangular inversion domain 95º–111ºW, 24º–39ºN. The state vector in this example has length $n = 235 + 8 = 243$.

155         The simplest (default) option for the user is to select a rectangular region of interest as latitude and longitude bounds. The IMI will then infer emissions for the 0.25° × 0.3125° grid cells within that region, excluding any grid cells more than 75% over water (adjustable default), and select 8 additional buffer elements with a k-means algorithm to pad out the rectangular inversion domain. The k-means algorithm sorts grid cells by latitude/longitude coordinates, and the number of buffer elements can be adjusted in the configuration file. Users also have the option to select an irregular region of interest, as in the Permian

example of Figure 1, by providing a previously defined state vector file or a shape file for the region boundaries.

**Table 1: Bottom-up methane emission inventories used as default prior estimates in IMI 1.0 [a]**

| Anthropogenic [b] | |
|---|---|
| United States | EPA GHGI (Maasakkers et al., 2016) [c] |
| Mexico | INECC (Scarpelli et al., 2020a) [d] |
| Canada | ECCC NIR (Scarpelli et al., 2022a) [e] |
| Rest of World | |
| Fuel exploitation | GFEI v2.0 (Scarpelli et al., 2022b) [f] |
| Other | EDGAR v6 (Janssens-Maenhout et al., 2019) [g] |
| **Natural** | |
| Wetlands | WetCHARTS v1.2.1 (Bloom et al., 2017) [h] |
| Geological seeps | Etiope et al., 2019 [i] |
| Open fires | GFED4 (Randerson et al., 2018) [j] |
| Termites | Fung et al., 1991 [k] |

[a] The inventories are archived on AWS on their native grids and over their temporal records, and are re-gridded and summed for use as IMI prior estimates through the HEMCO emissions processor in GEOS-Chem (Lin et al., 2021). The inventories listed here are those available

as of January 2022. They may be updated in the future as improved or more recent emission inventory data become available.

[b] All anthropogenic emissions are on a 0.1° × 0.1° grid and resolved by emission sector. They do not vary with time of year except for manure (Maasakkers et al., 2016) and rice (Zhang et al., 2021).

[c] Gridded version of the US EPA Inventory of US Greenhouse Gas Emissions and Sinks (GHGI; EPA, 2016) for 2012.

[d] Gridded version of the INECC national inventory (INECC and SEMARNAT, 2018) for 2015.

[e] Gridded version of the ECCC National Inventory Report (NIR; ECCC, 2020) for 2018.





[f] Global Fuel Emission Inventory (GFEI v2) constructed by gridding the national sectoral emission inventories reported by individual countries to the UNFCCC for 2018 and 2019.
[g] Data for 2018.
[h] Emissions for individual years and months specified on a 0.5° × 0.5° grid from the mean of the WetCHARTs ensemble.
[i] Scaled to a global total emission of 1.6 Tg a[-1] (Hmiel et al., 2020)
[j] Daily emissions specified on a 0.25° × 0.25° grid from the Global Fire Emissions Database (GFED4).
[k] Emissions specified on a 4° × 5° grid.

## 2.4 Prior emission estimates

The prior emission estimates $x_a$ should represent the best knowledge of methane emissions prior to performing the inversion. They need to be available in gridded format to match the resolution of the inversion. Table 1 compiles the bottom-up emission inventories used as default prior estimates in the IMI. The North American anthropogenic emissions are gridded versions of the national sector-resolved inventories reported by the individual countries to the United Nations Framework Convention on Climate Change (UNFCCC) as given by Maasakkers et al. (2016) for the United States, Scarpelli et al. (2020a) for Mexico,

and Scarpelli et al. (2022a) for Canada. The emissions from fuel exploitation (oil, gas, coal) in the rest of the world similarly grid the national emissions reported to the UNFCCC (Scarpelli et al., 2022b). The Emission Database for Global Atmospheric Research (EDGAR) v6 is otherwise used as global default. Natural emissions include contributions from wetlands with monthly resolution (Bloom et al., 2017), open fires with daily resolution (Randerson et al., 2019), and small sources from geological seeps and termites. These default inventories can be superseded by users with their own prior estimates, and we

give an example of this in Section 4.

The inversion infers emissions on the 0.25° × 0.3125° grid and this may include contributions from different sectors. Users can attribute the corrections to individual sectors based on the sectoral distribution of the emissions in the prior inventories and estimates of prior errors for each sector (Shen et al., 2021; Cusworth et al., 2021a). This needs to be done in post-processing of the inversion results.

## 2.5 TROPOMI operator


The forward model $y = F(x)$ for the inversion involves successive application of a GEOS-Chem operator $C = G(x)$ that relates the emission state vector $x$ to the resulting 3-D simulated dry-air mixing-ratio field $C$, and a TROPOMI operator $y = T(C)$ that relates the vertical profile of simulated dry-air mixing ratios to the corresponding column-average dry-air mixing ratio ($X_{CH4}$) that would be reported by TROPOMI. The TROPOMI retrieval provides information on the operator $T$ as the

dependence of $X_{CH4}$ on the local vertical profile vector of dry-air mixing ratios $c$ (with prior estimate $c_A$) for 12 sub-column pressure layers extending from the local surface to the top of the atmosphere, with vertical sensitivity described by a column averaging kernel vector $a$ for those 12 layers:

$$X_{CH4} = a^T c + (1 - a)^T c_A, \tag{1}$$





where **1** denotes a 12-dimensional vector of unit values.

Figure 2 summarizes the operations involved in simulating TROPOMI observations of the GEOS-Chem atmosphere. The first step is to geo-locate the TROPOMI pixel (nadir resolution $5.5 \times 7$ km$^2$, but coarser off-nadir) on the GEOS-Chem $0.25° \times 0.3125°$ grid. If the pixel overlaps two or more GEOS-Chem grid cells then the calculation is done for each grid-cell column followed by area-weighted averaging. We map the sub-column mixing ratios from the GEOS-Chem vertical grid (47 layers) to the TROPOMI vertical grid (12 layers) with partial allocation of GEOS-Chem layers to TROPOMI layers on the
basis of pressure edges (Figure 2). We then apply the TROPOMI column averaging kernel vector $\boldsymbol{a}$ with equation (1) to obtain the column-average dry-air mixing ratio $X_{CH4}$ as would be observed by TROPOMI in the GEOS-Chem atmosphere. We address differences in surface pressure between GEOS-Chem and TROPOMI by applying the lowest-level sub-column mixing ratio in GEOS-Chem down to the lowest TROPOMI pressure edge, as illustrated in Figure 2.

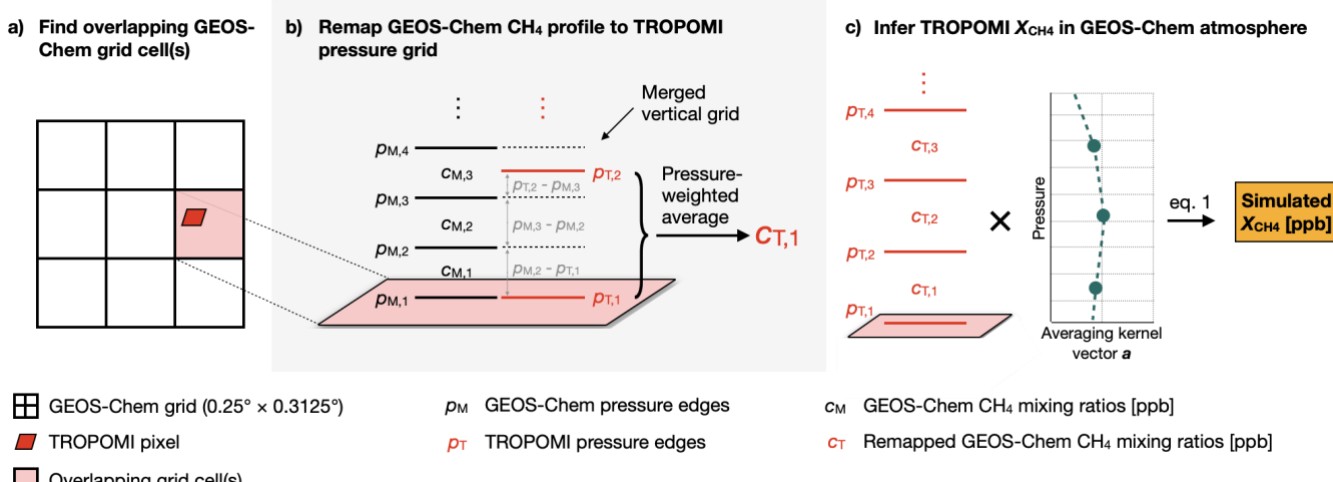

**Figure 2:** Simulation of TROPOMI column-average dry-air mixing ratio ($X_{CH4}$) observations in the GEOS-Chem 3-D model atmosphere. a) The operator first identifies which GEOS-Chem grid cells overlap with the TROPOMI observation pixel. b) The operator remaps conservatively the GEOS-Chem vertical profile of methane dry sub-column mixing ratios $\boldsymbol{c_M}$ from the GEOS-Chem pressure grid $\boldsymbol{p_M}$ to the TROPOMI pressure grid $\boldsymbol{p_T}$ to produce a vertical profile of methane sub-column mixing ratios $\boldsymbol{c_T}$ on the TROPOMI pressure grid. c) The
TROPOMI averaging kernel vector $\boldsymbol{a}$ (equation 1) is applied to the remapped GEOS-Chem profile on the TROPOMI pressure grid to produce a virtual $X_{CH4}$ observation of the GEOS-Chem atmosphere. If multiple GEOS-Chem grid cells overlap with the TROPOMI observation, the corresponding $X_{CH4}$ values are area-weighted to the TROPOMI pixel.

    The column averaging kernel sensitivities in TROPOMI are generally within two percent of unity in the troposphere
and drop off slowly in the stratosphere (Hu et al., 2016). Thus the pressure remapping has relatively little effect except in regions with strong topography, where high-elevation pixels have greater stratospheric contribution to $X_{CH4}$. Stratospheric methane in GEOS-Chem is unbiased except under polar vortex conditions (Zhang et al., 2020), so the IMI can account for these elevation changes.





## 2.6 Optimization procedure

Our Bayesian inversion to infer methane emissions fits the GEOS-Chem simulation to the TROPOMI observations, weighing prior and observational uncertainties and assuming normal error pdfs. This involves minimization of the scalar cost function (Brasseur and Jacob, 2017)

$$J(x) = (x - x_A)^T S_A^{-1}(x - x_A) + \gamma(y - Kx)^T S_O^{-1}(y - Kx),$$ (2)

where $K = \partial y/\partial x$ is the Jacobian matrix, $S_A$ is the prior error covariance matrix, $S_O$ is the observational error covariance
matrix including contributions from instrument and forward model errors, and $\gamma$ is an additional regularization parameter. $K$ describes the sensitivity of observations $y$ to the state vector $x$ as described by the forward model $F(x)$. It is computed column by column from an ensemble of perturbation simulations in the forward model, each perturbing a single element of the state vector from the reference simulation. Because the model is strictly linear, $K$ defines GEOS-Chem for the purpose of the inversion.

240          The default $S_A$ is constructed in the IMI by assuming 50% error standard deviation on emissions, with no error correlations (diagonal matrix). The default $S_O$ assumes a uniform observational error standard deviation of 15 ppb, based on previous estimates of 13-15 ppb for TROPOMI by the residual error method (Qu et al., 2021; Shen et al., 2021), again with no error correlation. These default values are adjustable by the user through the configuration file.

             The regularization parameter $\gamma$ is used to prevent overfitting/underfitting that would result from inexact specifications
of $S_A$ and $S_O$, and because the observations are not perfectly independent and identically distributed (IID condition). The best value for $\gamma$ can be selected on the basis of the L-curve (Hansen et al., 1999) or the expected Chi-square distribution of the cost function's prior terms (Lu et al., 2021). These two methods yield consistent results (Qu et al., 2021). Shen et al. (2021) used the L-curve to select $\gamma = 0.25$ for a regional inversion of TROPOMI observations over eastern Mexico at $0.25° \times 0.3125°$ resolution. We adopt that value in the IMI as default but it can be adjusted in configuration.

250          The posterior state vector $\hat{x}$ minimizing $J(x)$ is obtained by analytical solution of $dJ/dx = 0$ as

$$\hat{x} = x_A + (\gamma K^T S_O^{-1} K + S_A^{-1})^{-1} \gamma K^T S_O^{-1}(y - Kx_A),$$ (3)

with posterior error covariance matrix (characterizing uncertainty in $\hat{x}$) given by

$$\hat{S} = (\gamma K^T S_O^{-1} K + S_A^{-1})^{-1}.$$ (4)

$\hat{S}$ provides full closed-form characterization of the error on $\hat{x}$ assuming that the inverse problem has been well posed through
the formulation of the cost function. Errors in the formulation of the cost function can be evaluated through an inversion ensemble varying inversion parameters, prior emission estimates, and satellite observation sampling. The averaging kernel matrix

$$A = I_n - \hat{S}S_A^{-1}$$ (5)


describes the sensitivity of $\hat{x}$ to the truth (i.e., $A = \partial\hat{x}/\partial x$). The trace of $A$, referred to as the degrees of freedom for signal
(DOFS; Rodgers 2000), measures the information content of the observations towards optimizing the state vector. It represents
the number of independent pieces of information on the state vector that the observations can quantify. The diagonal entries
of $A$ are referred to as averaging kernel sensitivities, and they give an estimate of how much the posterior solution for a given
state vector element is informed by the observations as opposed to the prior estimates (Cui et al., 2014; Brasseur and Jacob,
2017). An emission element with averaging kernel sensitivity 0 is not quantified by the observations at all, and the inversion
results for that grid cell return the prior value. An emission element with averaging kernel sensitivity 1 is fully quantified by
the observations, and the inversion results for that grid cell are independent of the prior estimate.

**2.7 IMI Preview: assessing information content before performing an inversion**

The IMI includes a preview feature designed to help users avoid expending resources on inversions with insufficient
information content. Lack of information could come from low TROPOMI data density (e.g., from cloud cover) and/or from
seriously biased prior emission estimates for the region/period of interest. The preview can be run after configuring the IMI
and before initiating the inversion, and it performs several tasks. First, it maps the TROPOMI data and prior emission estimates
for the selected region and period of interest, so the user can assess spatial correspondence between the two datasets. Second,
it maps observation density and counts the total number of observations available for the selected region and period. Third, it
maps retrieved SWIR albedo across the scene, to help users identify potential artifacts in the methane retrieval if the SWIR
albedo retrieval shows similar features (Barré et al., 2021). Fourth, it estimates the USD financial cost of performing the
inversion by scaling the cost of our illustrative Permian Basin inversion (Section 4.3) according to the number of state variables,
grid resolution, and inversion period length. Finally, it makes a rough estimate of the expected DOFS for the user's inversion
using the procedure outlined below. A detailed example of the IMI preview feature is presented in Section 4.2.

The rough estimate of the expected DOFS is done as follows. Ignoring error correlations, assuming uniform
observational errors, and further assuming uniform transport, the calculation of the averaging kernel matrix reduces to a scalar
problem (Brasseur and Jacob, 2017). The averaging kernel sensitivity $A$ for a single emission element in the state vector is
given by

$$A = \frac{\sigma_A{}^2}{\sigma_A{}^2 + \frac{(\sigma_O/k)^2}{m}},$$
(6)

where $\sigma_A$ (kg m$^{-2}$ s$^{-1}$) is the prior error standard deviation of the emission element, $\sigma_O$ (mol mol$^{-1}$) is the observational error
standard deviation, $m$ is the number of satellite observations relevant to that emission element, and the transport model is
defined by the parameter $k$ (m$^2$ s kg$^{-1}$) as a summary representation of the Jacobian. With default 50% prior error standard
deviation, we have $\sigma_A = 0.5\, Q_A/(nL^2)$, where $Q_A$ (kg s$^{-1}$) is the total prior emission for the region of interest, $n$ is the number
of emission elements in that region of interest (not counting the buffer elements), and $L$ (m) is the grid cell side-length (25 km
in the GEOS-FP default). For our guiding Permian Basin example using the default IMI emission inventories, $Q_A = 1.1$ Tg a$^-$





$^{1}$ and $n = 235$, which yields $\sigma_A = 1.2 \times 10^{-10}$ kg m$^{-2}$ s$^{-1}$. The number of observations $m$ per emission element is the total number of observations for the region and period of interest, divided by $n$; for the May 2018 Permian example we obtain $m = 86$ from 19,978 observations (see Section 4.2). $\sigma_O$ is by default $15 \times 10^{-9}$ mol mol$^{-1}$.

To estimate $k$ we use the approximation proposed by Nesser et al. (2021) for simple mass balance ventilation of local emissions in the grid cell by a constant wind:

$$k = \alpha \frac{M_{\text{air}}}{M_{\text{CH4}}} \frac{Lg}{Up}, \tag{7}$$

where $M_{\text{air}}$ is the molar mass of dry air, $M_{\text{CH4}}$ is the molar mass of methane, $g$ is gravitational acceleration, $U$ is a uniform wind speed ventilating the emission element (assumed 5 km h$^{-1}$), and $p$ is the surface pressure (assumed 1010 hPa). The parameter $\alpha$ serves as a simple representation of turbulent diffusion, and here we take $\alpha = 0.4$ following Nesser et al. (2021) so that $k = 1.26$ m$^2$ s kg$^{-1}$. After computing $A$ in this way, the expected information content for the inversion can be obtained

as

$$\text{DOFS} = nA = \frac{n\sigma_A{}^2}{\sigma_A{}^2 + \frac{(\sigma_O/k)^2}{m}} . \tag{8}$$

Equation (8) gives a quick estimate of the information content to be expected from the inversion without actually running the IMI. Although very rough, it is based on the same principles as the actual inversion and we find that it gives a successful approximation. It further has the advantage of being transparent in that $n$ and $m$ are defined by the user choice of

region and period of interest, $\sigma_A$ and $\sigma_O$ are set by default in the IMI but are configurable by the user, and $k$ has direct physical meaning. In fact, $k$ can be used for a very rough estimate of emissions corresponding to a local column enhancement (Jacob et al., 2016).

The user may decide on the basis of the DOFS estimated from equation (8) whether or not to carry out the inversion. DOFS ~ 1 would be a minimum requirement to achieve any solid information on emissions in the region of interest, and more

may be desirable if multiple pieces of information are desired on the emission fields within the region. Shen et al. (2022) required DOFS > 2 to reliably estimate emissions from oil/gas basins in North America. If the user deems the DOFS to be insufficient, a cure is to increase the number of observations by lengthening the observation period. The user may also revisit the information on the prior emission estimate and whether a larger value of $\sigma_A$ may be appropriate, which will increase the DOFS.

Beyond inspection of the DOFS, the user should inspect the preview plots to guard against large artifacts in the observations or large bias in the spatial distribution of prior estimates. Artifacts in the observations can be diagnosed by similarity of patterns between $X_{\text{CH4}}$ and SWIR albedo, implying that spectral features in the albedo are propagating into the $X_{\text{CH4}}$ retrieval. If so the observations should not be used. Large bias in the spatial distribution of prior estimates can be diagnosed by comparison to the TROPOMI observations, and would be problematic in the inversion by misallocating the

corrections (Yu et al., 2022); this can be addressed by increasing the error on the prior estimate, using a non-informative





uniform prior estimate, or switching to a different prior emission inventory, as will be illustrated in Section 4 in the context of the Permian example.

## 3 Implementation of the IMI on the cloud

Figure 3 outlines the architecture of the IMI on the AWS cloud including the preview and the inversion workflow. The IMI
draws on two AWS facilities: the Elastic Compute Cloud (EC2) for computation and the Simple Storage Service (S3) for data storage. The computing environment for the workflow is contained in an Amazon Machine Image (AMI) accessible from the EC2 service. The TROPOMI operational data are archived independently in their own S3 bucket by MEEO. Meteorological data from the NASA Goddard Earth Observing System Fast Processing (GEOS-FP) product are archived in another S3 bucket to support GEOS-Chem on the cloud (Zhuang et al., 2019). That bucket also contains the bottom-up methane emission
inventories that serve as default prior estimates for the inversions (Table 1). Smoothed TROPOMI data serving as boundary conditions for the inversions have their own S3 bucket. All of these datasets are accessed by the preview and the workflow as needed, by automated transfer from S3 to the Elastic Block Storage (EBS) volume on the user's EC2 instance.
Engineering Company, Annandale, NJ, USA.

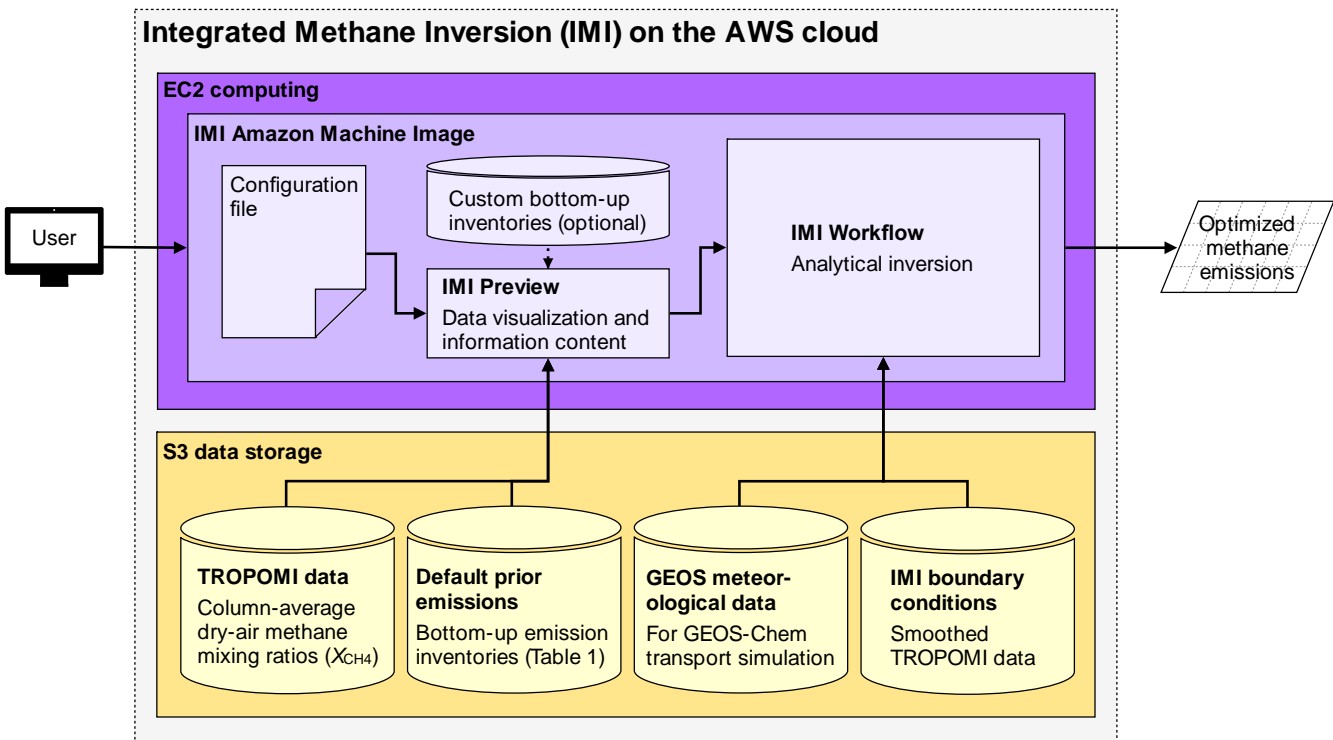


**Figure 3:** Integrated Methane Inversion (IMI) preview and workflow on the Amazon Web Services (AWS) cloud to infer methane emissions from TROPOMI data. The IMI is accessed as a custom Amazon Machine Image (AMI) on the AWS EC2 computing service. It





accesses the operational TROPOMI methane data, GEOS meteorological data, default bottom-up emission inventories, and IMI boundary conditions (smoothed TROPOMI data) from AWS S3 data storage buckets for the desired period. All of these data are resident on the cloud. Users specify their region/period of interest through a configuration file that also allows modification of IMI defaults. They can provide alternative bottom-up inventories (instead of the GEOS-Chem defaults) to serve as prior estimates for the inversion. The IMI preview provides visualization of the TROPOMI data and prior emission inventories, and a rough estimate of the information content of the inversion (degrees of freedom for signals or DOFS). Based on this information the user can decide to carry out the inversion through the IMI workflow (Figure 4) or modify the configuration (see Sections 2.7 and 4.2 for details).


Workflow users begin by opening an EC2 instance and selecting the workflow AMI. The AMI contains the GEOS-Chem and IMI source codes, a configuration file, and all required software dependencies. They then specify a region and time period of interest in the configuration file. The configuration file also contains options to modify the IMI default settings (Table 2). Detailed instructions for configuring the IMI are provided in the online technical documentation (imi.seas.harvard.edu). Users can use as prior estimates the default bottom-up emission inventories provided with the workflow (Table 1) or they can substitute their own. They can run the IMI preview (Figure 3) to collect and visualize the TROPOMI and prior emission data for their selected region and time period, and to get a rough estimate of information content and cost (Section 2.7). The preview incurs no significant computational cost. If the information content is deemed sufficient, the user can go on to run the IMI, including construction of the Jacobian matrix. This is the main computational cost but is very reasonable for typical inversion domains and periods (see Section 4.3). Once the Jacobian matrix has been constructed to define the forward model transport, it can be re-used to populate an inversion ensemble at no significant added computational cost by varying inversion parameters and/or bottom-up emission inventories (the latter requires rescaling the matrix). It can also be archived for later use.

**Table 2: Default IMI version 1.0 settings and configuration options.**

| Setting | Default | Configuration options |
|---|---|---|
| State vector | Rectangular domain [a] | Irregular domain [b] |
| Spatial resolution and meteorological data | $0.25° \times 0.3125°$ (GEOS-FP) | $0.5° \times 0.625°$ (MERRA-2) |
| Observational error standard deviation | 15 ppb | Any uniform value |
| Prior error standard deviation | 50% | Any uniform value |
| Regularization parameter $\gamma$ | 0.25 | Any value |
| Buffer zone width [c] | 5° | Any uniform value |
| Number of buffer elements in state vector | 8 | Any number [d] |
| Spin-up time | 1 month | Any length |

[a] Defined automatically from user-selected latitude and longitude bounds for the region of interest.

[b] Either specified with a shapefile or defined by a pre-generated custom state vector file.

[c] Extension of the inversion domain beyond the region of interest to absorb errors in boundary conditions.

[d] Buffer elements are specified with a *k*-means algorithm.



The current IMI version 1.0 limits the choice of regions to within North America (10°-70°N, 40°-140°W), Europe (33°-61°N, 30°W-70°E), and Asia (11°S-55°N, 60°E-150°E). This improves computational performance by allowing use of pre-cut continental subsets of the GEOS meteorological data corresponding to the default windows used in GEOS-Chem nested applications (Kim et al., 2015; Zhang et al., 2015). The meteorological data for these three windows are uploaded to AWS by the GEOS-Chem Support Team with a latency of a few weeks. Users may apply the IMI to other regions, but this requires

cropping global meteorological fields to a suitable nesting domain following instructions on the GEOS-Chem website (http://www.geos-chem.org). Future IMI versions will expand the pre-cut windows to other continents.



**Figure 4:** Flowchart for the Integrated Methane Inversion (IMI 1.0) on the AWS cloud. Here $\boldsymbol{x}$ is the emission state vector of length $\boldsymbol{n}$, $\boldsymbol{y}$ is the vector of TROPOMI observations, $\boldsymbol{C}$ is the time-evolving 3D GEOS-Chem methane concentration field over the inversion period, $\boldsymbol{\mathcal{G}}$ is GEOS-Chem, $\boldsymbol{\mathcal{T}}$ is the TROPOMI operator, $\boldsymbol{K}$ is the Jacobian matrix, $\widehat{\boldsymbol{S}}$ is the posterior error covariance matrix, and $\boldsymbol{A}$ is the averaging kernel matrix. See Section 2 for equations and further description of the algorithm. The workflow has the option of skipping the calculation of the Jacobian matrix $\boldsymbol{K}$ if it has already been computed; this allows generation of a solution ensemble by varying inversion parameters (see text for details).



Figure 4 charts the IMI computational workflow as described in Section 2 and contained in the AMI. The workflow receives instructions from the configuration file and then has three basic steps: (i) perform an ensemble of GEOS-Chem simulations to define the transport features for individual emission state vector elements, (ii) use those simulations to construct the Jacobian matrix, and (iii) solve the analytical inversion using equations (3-5). When the user configures and runs the IMI,

these steps are executed automatically to generate posterior methane emission estimates for the inversion domain along with error statistics. The user can then inspect the inversion results using a visualization notebook provided with the IMI. The notebook contains sample code to plot the state vector, prior emissions, posterior emissions, scale factors (posterior/prior ratios), averaging kernel sensitivities, and TROPOMI data for the inversion domain and period.

The IMI workflow begins by constructing the emission state vector (length $n$) from the user specifications. After an
initial spin-up simulation to generate initial conditions for the period of interest, it then performs $n + 1$ GEOS-Chem simulations. These include a reference simulation driven by the prior bottom-up emission inventories and $n$ perturbation simulations perturbing one emission element at a time. All of these simulations access S3 data for prior emissions, meteorology, and boundary conditions (Figure 3). The perturbation simulations determine the sensitivities of the satellite observations to the state variables and are used to construct the Jacobian matrix $K$ as described in Section 2.6. For our one-month Permian basin
example ($n = 243$), a total of 244 simulations are performed in this way. The reference and perturbation simulations are embarrassingly parallel and can be performed simultaneously once the spin-up simulation is complete if $n + 1$ CPUs are available on the user's EC2 instance; with fewer CPUs the workflow runs the simulations in parallel batches.

After computing $K$ from the reference and perturbation simulations, the IMI solves equations (3-5) for the optimized emission estimates $\hat{x}$, posterior errors $\hat{S}$, and averaging kernel matrix $A$, and saves these quantities as output. The elements of
$\hat{x}$ and the diagonal entries of $A$ (averaging kernel sensitivities) and $\hat{S}$ are then mapped to the grid cells of the inversion domain and saved as a separate output to facilitate inspection of the results. The final step of the workflow is to conduct a GEOS-Chem simulation using the posterior emissions $\hat{x}$ for comparison to the TROPOMI observations and to a GEOS-Chem simulation using prior emissions (reference simulation), to verify the quality of the inversion results in better fitting the TROPOMI observations. Results are provided as part of the IMI output.

**4 Illustrative application to the Permian Basin**

**4.1 Setup**

We perform a one-month inversion for the Permian Basin (currently the most prolific US oil-producing basin) as an illustrative application of the IMI. We choose a 1-month period (1-31 May 2018) for the inversion. The region of interest is defined from a shapefile for the Permian Basin and comprises 235 state vector elements to describe emissions within the region at 0.25° ×
0.3125° resolution, plus 8 buffer elements to pad out the inversion domain, for a total of 243 state vector elements (Figure 1).





We perform the inversion using the default IMI settings laid out in Tables 1 and 2 but with the custom state vector of Figure 1. The steps prior to initiating the workflow are as follows:

1. Create an AWS instance with the IMI workflow AMI.
2. Connect to the instance, upload the custom state vector file of Figure 1, and open the configuration file.
3. Set the start date to 1 May 2018 and the end date to 1 June 2018.
4. Turn off the option to automatically generate the state vector from the latitude and longitude bounds of a rectangular region of interest.
5. Enter the path to the custom state vector file and close the configuration file.
6. Run the IMI preview to display the TROPOMI data and prior emissions, and estimate the information content to be achieved in the inversion.

## 4.2 Analysis of results

Figure 5 shows the IMI preview results including the mean TROPOMI $X_{CH4}$ data for the selected region and period, the observation density, the retrieved SWIR albedo, and the default prior emission estimates (EPA GHGI). The TROPOMI data (Fig. 5a) include $N = 19,978$ individual observations for the region of interest and these are used for the DOFS estimate in the preview. There are more than 100,000 additional observations in the inversion domain outside the region of interest and covering the buffer grid cells (Figure 1). The two methane hotspots at centre-image correspond to the Permian's Delaware and Midland sub-basins. TROPOMI provides relatively uniform sampling across the region of interest (Fig. 5c), and Figures 5a and 5d show no indication of albedo-related regional $X_{CH4}$ biases. However, we see that the gridded GHGI inventory (Fig. 5b) misrepresents the spatial distribution of emissions in the Permian by failing to capture the sub-basin structure apparent in Fig. 5a. Furthermore, the inversion preview indicates an expected DOFS of 2.0, which is marginal for quantifying emissions on that regional scale (Shen et al., 2021b).

At this point it would be sensible to reconfigure the IMI before performing the inversion, and we will explain how in what follows. If we proceed and conduct the inversion with these default settings, we find a DOFS of 1.9 (close to the preview). The posterior emission integrated over the region of interest is 1.8 Tg a$^{-1}$, much higher than the default GHGI prior emission of 1.1 Tg a$^{-1}$, and with scale factors (posterior/prior ratios) ranging from 1.0 to 2.7. These results are consistent with independent observations that the GHGI emissions for the Permian Basin are far too low (Omara et al., 2018; Robertson et al., 2020; Chen et al., 2021; Cusworth et al., 2021b; Irakulis-Loitxate et al., 2021; Lyon et al. 2021), but the low DOFS and biased spatial distribution in the prior emissions do not inspire confidence in the results.

One can increase the DOFS simply by increasing the length of the inversion period, thus accumulating more observations, but the incorrect spatial distribution of the prior estimate makes it much harder for the inversion to converge to the correct solution (Yu et al., 2021). This could be addressed in configuration by increasing the magnitude of the prior error estimate, or by replacing the default prior emission inventory with a non-informative prior of uniform emissions. These ad hoc



adjustments may provide instructive results, though increasing prior errors may result in unphysical solutions if the problem
is underconstrained in part of the domain, as is typical for satellite-based inversions.


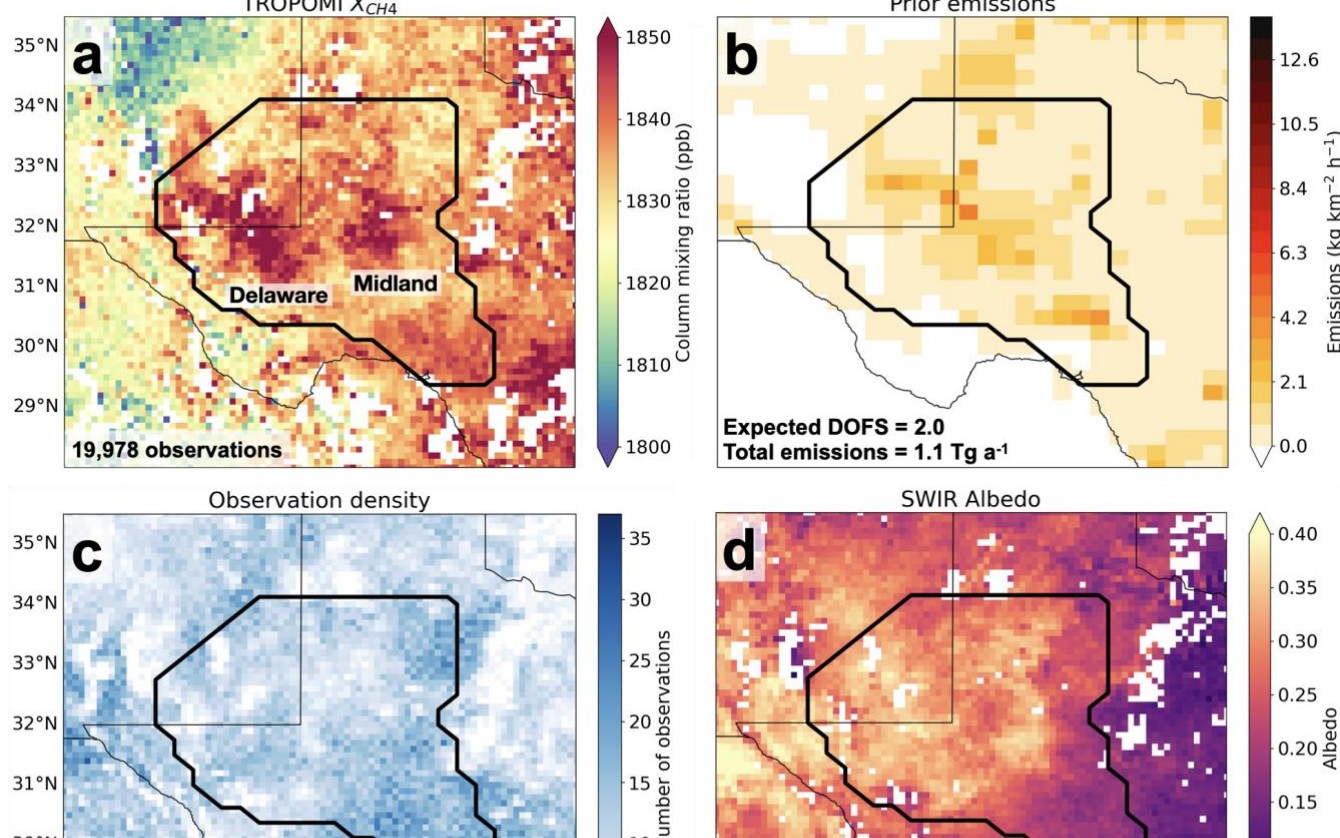

**Figure 5:** Output of the IMI preview (Section 2.7) applied to the Permian Basin example with the default EPA gridded GHGI inventory
(Maasakkers et al., 2016) as prior estimate of emissions. (a) Mean TROPOMI column-average dry-air methane mixing ratio ($X_{CH4}$) data for
the user-selected region (thick black contour) and period of interest (1-31 May 2018), resampled to a 0.1° × 0.1° grid and cropped to 98.5°W–
107°W, 28°N–35.5°N for visibility. The colour bar is saturated to highlight methane hotspots over the Delaware and Midland sub-basins.
Inset gives the total number of observations and degrees of freedom for signal (DOFS) for the region of interest. (b) Gridded GHGI (default)
prior emissions. (c) Number of observations per 0.1° × 0.1° grid cell for the period of interest. (d) Mean SWIR albedo for the period of
interest on the 0.1° × 0.1° grid. Here the preview shows poor agreement in the spatial distribution of emissions between the observations and
prior emission estimates, suggesting that the prior estimate should be replaced by a better one (as is done in our application) or that the prior
error estimate should be increased.

A better alternative is to investigate whether an improved bottom-up inventory would enable a more accurate
inversion. In the case of the Permian Basin, a better gridded bottom-up inventory is available from the Environmental Defense





Fund (EDF) with more accurate accounting of oil and gas infrastructure and larger total emissions of 2.7 Tg a$^{-1}$ (Zhang et al.,
2020). IMI results using the EDF inventory as custom bottom-up prior estimate are shown in Figure 6. Starting with the IMI

preview, we find that the spatial distribution of prior emissions is much more consistent with the TROPOMI data (Figure 6a,

compare to Figure 5b), with much higher expected DOFS of 11.7 that reflects the higher prior emissions (and hence the larger

absolute prior error standard deviations). Proceeding to run the IMI workflow, we find that the posterior emissions now total

3.5 Tg a$^{-1}$, up 30% from the prior estimate of 2.7 Tg a$^{-1}$ and with clear demarcation of the two sub-basins. The new scale factors

range from 0.59 to 2.20, reflecting a need for both increased and decreased emissions in different parts of the basin to better

match the satellite data. The averaging kernel sensitivities yield a DOFS of 10.8 (consistent with the IMI preview), which gives

us confidence in the inversion results both on the basin scale and in the spatial allocation within the basin. In particular, we

see the need for larger increase of emissions in the Midland than Delaware sub-basin.


**Figure 6:** Results of a one-month (1-31 May 2018) application of the IMI to the Permian Basin using the EDF emission inventory (Zhang et al., 2020) as prior estimate of emissions. (a) Prior emissions. (b) Posterior emissions. (c) Scale factors applied to the prior emissions to obtain the posterior emissions. (d) Averaging kernel sensitivities with associated degrees of freedom for signal (DOFS) inset.





Figure 7 shows the GEOS-Chem simulations for the inversion period with the prior and posterior emissions. The

posterior simulation produces much higher methane concentrations over the Midland sub-basin, better matching the

TROPOMI observations of Figure 5. The mean GEOS-Chem-TROPOMI bias across the region of interest improves from -2.2

ppb in the reference simulation to -1.5 ppb in the posterior simulation, and the Pearson correlation coefficient improves from

0.22 to 0.37.

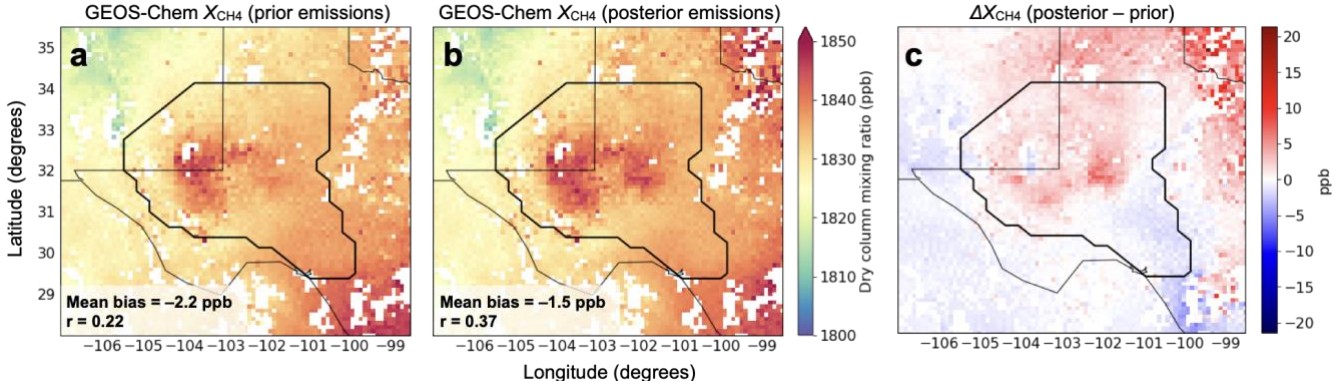

**Figure 7:** GEOS-Chem simulations of TROPOMI $X_{CH4}$ observations for May 2018 with (a) prior emissions and (b) posterior emissions. Panel (c) shows the difference between the two. The contour line shows the Permian Basin selected as region of interest in the inversion. The insets give the mean bias and Pearson correlation coefficient for the region of interest in comparison to the TROPOMI observations in Figure 5a.

## 4.3 Cost

We conducted the illustrative inversion presented here on an AWS EC2 c5.9xlarge instance with 36 CPUs and 500

GB of EBS storage. Compute wall-time was 10 hours, with the bulk of that time spent constructing the Jacobian matrix $K$.

Our cost was $20 USD for an 'on-demand' instance, in which the requested resources are made available almost immediately.

A 1-year inversion would cost about $330 USD (12 × $20 = $240, plus the cost of additional EBS storage to accommodate the

longer inversion period), and wall-time could be reduced by requesting more CPUs at no additional cost since the charge is

per CPU-hour. Performing additional inversions with different parameters and prior inventories (Table 2, Section 4.2) adds

little cost because there is no need to reconstruct $K$. Data download and transfer between AWS services may incur some cost

but this is also minimal. A cheaper alternative to on-demand instances are "spot instances", which tap unused EC2 capacity

and can reduce costs by a factor of 3-4 or more (Zhuang et al., 2019). Spot instances can be reclaimed by AWS at any time,

which would cause the IMI to crash, but in practice this is rare and users can generally expect to retain a spot instance for up

to a month of wall-time (Pary, 2018).

## 5 Conclusions and future developments

There is a growing demand for tools to infer regional methane emissions with high resolution from satellite data. Our

Integrated Methane Inversion (IMI) workflow addresses this demand by enabling researchers and stakeholders to estimate



methane emissions for regions of interest at 0.25° × 0.3125° (≈ 25 × 25 km$^2$) resolution by Bayesian inversion of TROPOMI
satellite observations on the AWS cloud, using cutting-edge inversion methodology and without requiring massive data
download or technical expertise. The workflow interfaces with TROPOMI operational data and the GEOS-Chem model
already resident on AWS. It makes use of bottom-up emission inventories, GEOS-FP meteorological data, and boundary
conditions (smoothed 3-D TROPOMI fields) that are also stored on AWS. There is no need for large TROPOMI data
download. By automatically accessing all the needed resources on the cloud, the IMI embodies the new paradigm of "bringing
compute to data" when working with very large datasets.

We outlined how users can configure and run the workflow to optimize methane emissions for a selected region and
period of interest. The configuration can be as simple as defining the region (latitude/longitude bounds) and time period
(start/end dates), or more complex for users wishing to customize different aspects of the inversion such as the state vector,
the prior and observational errors, or the emission inventories used as prior estimates. The TROPOMI and GEOS-FP data are
operationally uploaded to the AWS cloud with a latency of a few days, so that continued access to current conditions is
available.

The inversion uses an advanced research-grade algorithm to derive best posterior estimates of emissions on the 0.25°
× 0.3125° grid by analytical solution to a Bayesian cost function. The analytical solution provides closed-form error statistics
on the posterior estimates and metrics on the information content from the observations including averaging kernel sensitivities
and the degrees of freedom for signal (DOFS). It enables no-cost error analysis by producing an ensemble of solutions
exploring the sensitivity to inversion parameters. The algorithm is fully documented in the literature (Turner et al., 2015;
Maasakkers et al., 2019, 2021; Zhang et al., 2021; Lu et al., 2022) including applications to TROPOMI data (Zhang et al.,
2020; Qu et al., 2021; Shen et al., 2021; 2022).

An IMI preview feature allows users to inspect the TROPOMI data and the anticipated quality of the inversion results
for the region and period of interest before committing to the actual inversion. The IMI preview inspects the TROPOMI data
for common artifacts correlated with SWIR albedo, determines the observation density across the region of interest, gives a
rough estimate of the DOFS to be expected from the inversion, and compares the spatial distribution of the prior estimates to
the TROPOMI data. Large differences in spatial distributions may require adjustments to the prior estimates for a successful
inversion.

We presented an illustrative application of the IMI workflow to a one-month inversion of TROPOMI observations
over the US Permian Basin. We showed how the DOFS and spatial distribution of prior emissions generated by the IMI preview
allowed us to identify the limitations of the initially intended first inversion, which we fixed by swapping in an improved prior
emission inventory. The subsequent inversion was performed at a cost of $20 USD using an AWS c5.9xlarge "on-demand"
instance with 36 CPUs, and could have been a factor of 3-4 cheaper using a "spot" instance.

This initial version of the IMI (version 1.0) has some limitations in functionality and does not include some of the
newer capabilities within the analytical inversion framework. Priority developments for future IMI versions include (1)
extension of pre-cut GEOS windows to continental domains outside of North America, Europe, and Asia; (2) option to use



lognormal rather than normal error pdfs for prior emissions to resolve the high tail of the emission distribution (Maasakkers et al., 2019; Lu et al., 2022); (3) option to use non-uniform prior and observational error covariance matrices; (4) more optimal

selection of state vector elements with a Gaussian mixture model (Turner and Jacob, 2015); (5) use of Kalman Filter techniques for continuous emission monitoring (Varon et al., 2022); (6) incorporation of data from future global-surveying satellite instruments including GeoCarb (Moore et al., 2018), CO2M (Sierk et al., 2019), MethaneSAT (Wofsy and Hamburg, 2019), and GOSAT-GW (Kasahara et al., 2020); and (7) application to inversions for CO and $CO_2$ emissions. This together with continued improvements to the operational TROPOMI methane product will make the IMI an increasingly powerful tool for

researchers and stakeholders to monitor methane emissions worldwide at high resolution using satellite data.

**Code availability**

Source code and documentation for the IMI are available at imi.seas.harvard.edu. The code used in this paper is permanently archived at https://doi.org/10.5281/zenodo.6081934.

**Author contribution**

DJV, MS, and DJJ contributed to the study conceptualization. DJV, MS, LE, WBD, LS, and SEH developed the model code. All authors contributed to the methods development. DJV performed the data analysis. DJV wrote the original draft and all authors reviewed and edited the manuscript..

**Competing interests**

The authors declare that they have no conflict of interest.

**Acknowledgements**

This work was supported by the ExxonMobil Research and Engineering Company and by the NASA Carbon Monitoring System.



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
