# Peer review of "Integrated Methane Inversion (IMI 1.0): A user-friendly, cloud-based facility for inferring high-resolution methane emissions from TROPOMI satellite observations"

_Geoscientific Model Development, 2022_

## Author Response (AR1)

**Responses to reviewers**

We thank the reviewers for their helpful comments and suggestions. Each of their comments is copied below in blue, and our responses follow in black.

In addition to the reviewer comments, we also addressed two minor errors in the original results of the 1-month demonstration inversion for the Permian Basin. First, we corrected an error that caused some of the bottom-up emissions in Table 1 to be loaded incorrectly by GEOS-Chem. The posterior emissions for the region and period are now 3.9 Tg/y rather than 3.5 Tg/y. Second, we corrected an error in the selection of data within the region of interest in Figure 7, and updated the reported error statistics accordingly.

**Reviewer #1**

1.  P2. L32: *"...is a major focus of climate policy"*
    Whose policy? Perhaps give an example. E.g. the COP26 global methane pledge.

    Good point and thank you for the suggestion. We have added a reference to the COP26 Global Methane Pledge.

2.  P2. L42: *It exemplifies the emerging concept of "bringing compute to data".*
    I think it would help to explain this jargon. At first glance it just looks like whoever came up with it doesn't know the difference between a noun and a verb.

    Agreed that the phrasing is a bit awkward, but it's common parlance in cloud-computing. We believe the quotation marks adequately set the phrase apart as a colloquialism and something that interested readers can search for on Google (search results).

3.  P2. L47: *"were limited by...sparse sampling (GOSAT)"*
    GOSAT is still active (along with GOSAT2) so it sounds odd to refer to it in the past tense like this.

    Thank you for catching this. We now note the years of activity for GOSAT and SCIAMACHY.

4.  P2. L62 *"Analytical solution takes advantage of the linearity of the relationship between methane emissions and concentrations (Maasakkers et al. 2021)"*
    This seems a peculiar citation. I hardly think that a paper from last year was the first to exploit this linearity. Analytical methane inversions date back several decades.

    Good point. We chose the Maasakkers et al. (2021) paper as a reference for the methane inversion problem being effectively linear despite dependence of the methane lifetime on OH, as those authors discussed that point in depth. We now add Chen and Prinn (2006) as an earlier reference for analytical methane inversions, followed by the Maasakkers paper for additional context.

5. P3. L69-70: *"and for stakeholders lacking technical expertise."*
   Who are these stakeholders? And isn't some technical expertise still required to interpret the results? If you don't understand how or why the results are being generated isn't there a danger that you might over-interpret the results without accounting for the many limitations of the satellite data, model and statistical analysis?

   Thank you for raising this question. Users need sufficient technical skills to run the code, follow the documentation, and understand the manuscript – but not the expertise required to build a satellite/methane inversion system from scratch. As with any open-access scientific software, misuse and poor interpretation of results are possible. One of our objectives in this paper is to exemplify practical use of the IMI. We now say so explicitly at the end of the introduction.

6. P4. L95, "in a public S3 bucket"
   This is just jargon. What does it mean?

   True! We removed the unnecessary mention of S3 buckets here. That terminology is explained later in the manuscript.

7. P4. L105: *"caused mainly by aliasing of surface albedo errors into the methane retrieval"*
   Are there other potential causes of regional biases beyond albedo?

   Lorente et al. (2021) identify surface albedo as the primary cause of regional biases, but biases can also arise from scattering-induced albedo errors and errors in surface altitude. We clarify this point.

8. P4. L110: *"and with SWIR albedo < 0.05"*
   I assume this should say ">0.05"? Otherwise you'd be filtering out almost all data.

   Thank you for catching this error!

9. P4. L110-112: What percentage of "good quality" observations do the additional filters remove?

   We now give the range of observations removed by the additional filters for North America in 2019.

10. P4. L124: *"and distributed vertically following a GEOS-Chem simulation at 4x5"*
    Why use the 4x5 simulation when 2x2.5 is available and might produce more accurate fields?

    Only for computational expediency, but upgrading to 2x2.5 boundary condition fields would be a good improvement for a future version of the IMI. We now include this as a priority development in the conclusions.

11. P5. L137: *"and the sinks are relatively smooth"*
    Do you mean spatially smooth? And is this referring to the offline sink fields in the model?

    Yes, we now specify "spatially smooth". We also now state that the sinks are from offline fields.

12. P5. Fig 1: This probably doesn't matter for this example work but is there an option for the buffer regions to include oceans? If your domain were further east then presumably you would need to somehow to account for oil and gas fields in the Gulf of Mexico within the buffer regions. Is that also covered by the 75% adjustable default?

    Users can set a lower land-cover fraction, down to 0%, to include ocean cells in the state vector, or manually edit the state vector file. We now state this in the text (Section 2.3) and add a row to Table 2 showing the minimum landcover fraction as an adjustable setting with default value 0.25.

13. P8. L206: *"The first step is to geo-locate the TROPOMI pixel…"*
    Are the TROPOMI data geo-located only over the 0.25x0.3125 inversion domain of interest (i.e. the Permian basin in this work) or over the full computational domain or some other spatial domain? If the first, is there a danger that important signals downwind of the region of interest are missed?

    The TROPOMI data are geo-located on the full inversion domain, including the region of interest and the surrounding buffer elements. We clarify this in the text (Section 2.5).

14. P8. L211-213: I'm not sure the figure is entirely clear. Do you prescribe the model surface pressure as the satellite surface pressure or the opposite way round? Does this run the risk of not conserving mass in the model column?

    The TROPOMI columns do not impact the model columns, so there is no risk of not conserving mass in the simulation. We rephrase for clarity.

15. P8. L227: *"…except under polar vortex conditions (Zhang et al 2020)."*
    This reference is surely wrong. Stanevich et al (2020) were the ones to show the nature of the GEOS-Chem latitudinal bias and errors were not entirely limited to polar vortex conditions. This leads back to the earlier point of why use the 4x5 simulation for boundary conditions rather than 2x2.5?

    Thank you for pointing this out. We now cite Stanevich et al. (2020) for the GEOS-Chem stratospheric bias. We also correct our original citation as Zhang et al. (2021), who showed that the stratospheric bias is largely restricted to polar vortex conditions (we incorrectly cited Zhang et al., 2020 in the original manuscript).

    Good point again about 4x5 vs 2x2.5 resolution for the boundary conditions; we now include upgrading to the higher resolution as a priority development for a next version of

the IMI in the conclusions.

16. P9. L244-245: Does the regularization parameter essentially limit the resolution of inferred emissions? Couldn't one just achieve a similar effect by using coarser resolution grid cells (i.e. coarser resolution perturbations), thus reducing runtime and AWS costs? The choice of the regularization parameter seems highly arbitrary, would the intended non-expert user understand this?

The regularization parameter has no impact on the resolution of inferred emissions; it effectively increases the observational error standard deviation. We explain in Section 2.7 that users can test different values for the regularization parameter via an inversion ensemble. We now mention gamma specifically as an inversion parameter users can vary when constructing an ensemble of inversion results (Section 2.6).

17. P11. L303-304: *"we find it gives a successful approximation"*
What counts as a successful approximation? It might be wise to show some evidence for this.

True. We now refer to Section 4.2, where we give two examples of expected and actual DOFS for the illustrative 1-month Permian Basin inversion with the default (gridded EPA) inventory and then with the EDF inventory.

18. P12. L321 *"using a non-informative uniform prior estimate"*
But isn't it the specification of Gaussian PDFs that enables one to perform a quick analytical inversion? Prescribing an alternative PDF would surely require an alternative inversion approach.

Thanks for catching this. We had in mind a Gaussian pdf with very large prior errors to serve as an effectively non-informative prior, but indeed that is not uniform. We rephrase for clarity.

19. P16. L401-404: Presumably if you have the Jacobian matrix already there's no need to perform a final forward GEOS-Chem simulation, since you can just take the product of x by K to generate modelled observations.

Thanks for pointing this out. It's true, we can correct the prior forward model results by $K(xhat - x_A)$. The advantage of running the posterior simulation is that it allows comparison with independent measurements, not just the TROPOMI data. We now explain this at the end of Section 3.

20. P18, L458: *"a better gridded bottom-up inventory"*
How do you know it is better? What has it been validated against?

We rephrase to call the EDF inventory an "alternative gridded bottom-up inventory".

21. P 20 Section 4.3
How do costs scale with domain size or number of state vector elements?

Thanks for bringing this up. We now explain that "Costs scale linearly with the area of the inversion domain and (for a fixed domain size) with the number of state vector elements".

22. P22. L534: *"option to use non-uniform prior and observation error covariance matrices"*
Does this also include considering the off-diagonal terms in the error covariance matrices?

Yes. We now say so explicitly.

23. P22. As it stands it appears there is only the option to run for a fixed time window with constant emissions assumed during this period (or at least a constant posterior scale factor applied to the temporally variant prior emission fields). Does the IMI have the capacity to run say a 1-year run resolving monthly emissions, or does each monthly run need to be conducted independently of others?

Each run needs to be conducted independently, but we plan to eventually release a Kalman filter mode that would enable the IMI to iteratively update emissions with a user-specified frequency (e.g., 1 week, 1 month). We now add this to the list of priority developments: "(5) use of Kalman Filter techniques for continuous emission monitoring with user-specified update frequency".

**Reviewer #2 – Christian Frankenberg**

1. Line 2: "resolution by inversion of satellite observations from the TROPOspheric" it reads is if observations are inverted (but fluxes are inverted so that the forward model matches observations). Maybe rephrase a bit.

This comment prompted us to review some of the atmospheric/geophysical inversion literature to identify the correct terminology – with somewhat surprising results. We found that while some authors refer to "data inversion" or "inversion of data" as we do (e.g., Stolt and Weglein, 1985; Eyre, 1999), others discuss "flux inversion" or "inversion of fluxes" (e.g., Deng et al., 2007; Schuh et al., 2010), and still others describe "forward model inversion" or more precisely "inversion of transport" (e.g., Gloor et al., 1999; Rödenbeck et al., 2003). This was instructive as it revealed a significant range of standard terminologies, with authors describing inverse analysis as inversion of every term in the y = F(x) inverse problem equation! We decided to rephrase "inversion of" as "inverse analysis of" in some places throughout the manuscript. In other places we retained the "data inversion" terminology.

2. Line 110: I am sure you meant to say albedo >0.05?

Yes, thank you for catching this error!

3. Line 149: While reading the manuscript, I was always waiting for the "time-dimension" of your state vector. Afterwards, I realized the setup uses one fixed time-period in which the emissions are supposed to be constant, right? This could be a week (with a low DOFs), a month (in your example) or a year or more I guess. Maybe it would be good to discuss the temporal aspect of the inversions up front somewhere, so that the general reader knows about it early on (or maybe it escaped me).

Thanks for pointing this out. We now explicitly state in the abstract, at the beginning of Section 2 (IMI methods), and in Section 2.3 (state vector definition) that the inversion is for period-average emissions.

4. Line 240: Is there an option to add co-variances to Sa? It seems an uncorrelated error per grid cell of 50% actually results in a rather low total prior error covariance for a regional aggregate. In fact, I am curious what the regional total prior uncertainty for your Permian run would be (you could add these prior uncertainties to the plots as well, e.g. Figure 6). Given that more naive users will use this inversion system, it could be very important to inform the user what the prior uncertainty of the target of interest total emissions are. People might otherwise naively assume it would also be 50% (which it would be with perfectly correlated errors). In your example inversion in Figure 3, your posterior is 30% higher than your prior. My guess is that the prior total uncertainty is quite a bit lower than 30%, given the amount of grid cells within the domain (I just wildly guesstimate it might be around 10% or lower).

Good point, we need to say that the assumption of uncorrelated errors leads to underestimated uncertainty in the region-wide total emission. We now acknowledge this in Section 2.6. We also now include treatment of error correlation as a priority development for a future version of the IMI in the conclusions.

5. Apart from that, well done, a good contribution to science, transparency, and a more "democratic" access to complex tools.

Thank you for your endorsement of our work!